# Numerical Study of Turbulent Air and Water Flows in a Nozzle Based on the Coanda Effect

**Youssef El Halal [1], Crístofer H. Marques [1], Luiz A. O. Rocha [2], Liércio A. Isoldi [1], Rafael de L. Lemos [1], Cristiano Fragassa [3] and Elizaldo D. dos Santos [1,*]**

[1]  School of Engineering, Universidade Federal do Rio Grande–FURG, Rio Grande, RS 96203-900, Brazil; youssefhalal20@gmail.com (Y.E.H.); cristoferhood@gmail.com (C.H.M.); liercioisoldi@furg.br (L.A.I.); er.lemos@outlook.com (R.d.L.L.)

[2]  Programa de Pós-Graduação em Engenharia Mecânica, Universidade do Vale do Rio dos Sinos (UNISINOS), São Leopoldo, RS 93022-750, Brazil; luizor@unisinos.br

[3]  Department of Industrial Engineering, Alma Mater Studiorum Università di Bologna, Viale del Risorgimento 2, 40136 Bologna, Italy; cristiano.fragassa@unibo.it

\*  Correspondence: elizaldosantos@furg.br; Tel.: +55-53-3233.6916

**Abstract:** In the present work it is performed a numerical study for simulation of turbulent air and water flows in a nozzle based on the Coanda effect named H.O.M.E.R. (High-Speed Orienting Momentum with Enhanced Reversibility). The main purposes of this work are the development of a numerical model for simulation of the main operational principle of the H.O.M.E.R. nozzle, verify the occurrence of the physical principle in a device using water as working fluid and generate theoretical recommendations about the influence of the difference of mass flow rate in two inlets and length of septum over the fluid dynamic behavior of water flow. The time-averaged conservation equations of mass and momentum are solved with the Finite Volume Method (FVM) and turbulence closure is tackled with the $k$-$\varepsilon$ model. Results for air flow show a good agreement with previous predictions in the literature. Moreover, it is also noticed that this main operational principle is promising for future applications in maneuverability and propulsion systems in marine applications. Results obtained here also show that water jets present higher deflection angles when compared with air jets, enhancing the capability of impose forces to achieve better maneuverability. Moreover, results indicated that the imposition of different mass flow rates in both inlets of the device, as well as central septum insertion have a strong influence over deflection angle of turbulent jet flow and velocity fields, indicating that these parameters can be important for maneuverability in marine applications.

**Keywords:** coanda effect; turbulence model; computational fluid dynamic; finite volume method; H.O.M.E.R. nozzle

## 1. Introduction

In 1936, Henri Coanda patented the first device capable to deflect a stream without any movable parts [1]. This phenomenon is named Coanda effect and consists on the tendency of a fluid to adhere to a curved surface due to the local pressure drop caused by acceleration of flow around a solid surface. Coanda surfaces are generated by asymmetric profiles configured in such as way that the flow properties in the exit of devices are changed [2]. One illustration of the domain subjected to Coanda effect flow can be seen in Figure 1. Thus, the Coanda effect became the target of many studies where several mechanisms which were based on this effect were developed, such as cooling [3–5], safety devices [6], aerospace applications [7–10] and few recent studies in marine devices [11,12]. The Coanda effect is also important for the design of Unmanned Aerial Vehicles (UAV). Some advantages on the use of this main operational principle for UAV as the capability to

control the direction of the net force and improvement of lift forces acting on the device have been reported in the literature [13–17].

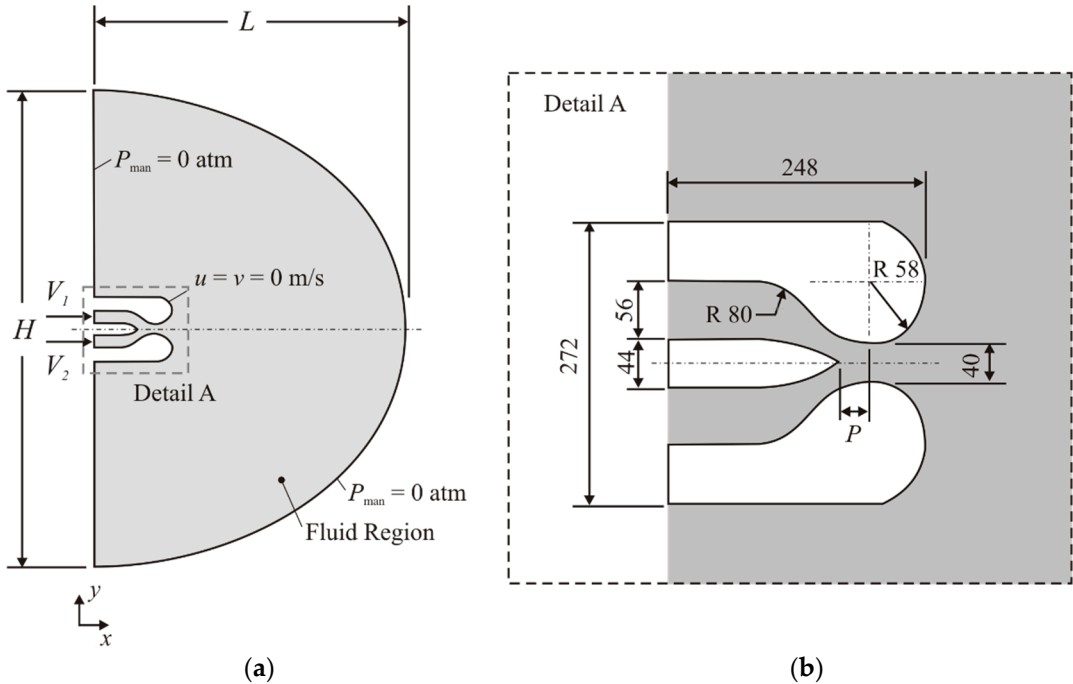

**Figure 1.** Computational domain of the studied problem: (**a**) entire domain and boundary conditions, (**b**) detailed view of the device (dimensions in mm).

Important works have been performed with the aim to improve the comprehension about the behavior of turbulent flows over aerodynamical profiles considering the Coanda effect. For instance, Ameri [18] conducted a theoretical and experimental study in ejectors with the aim to develop models for prediction of device performance based in this main operational principle. According to the author, in spite of large use of this principle in aeronautical applications, the challenge to understand the physical phenomenon leads to difficulties for design of the devices. Afterwards, Kim et al. [19] studied numerically the effect of some parameters, as pressure ratio between primary and secondary nozzles, over the performance of a Coanda nozzle. Djojodihardjo et al. [20] performed a numerical study using the Coanda effect applied to the control of wind turbines. More precisely, it is shown how the effect of a jet over the aerodynamical profile can be used to augment the lift and decrease the drag in the turbine blades. Gan et al. [21] simulated a two-dimensional jet flow over a logarithmic surface seeking to evaluate the behavior of the oncoming jet over the profile. Moreover, a parametric study about the effect of the outlet height of device over fluid dynamic behavior of jet flow was performed. Afterwards, Trancossi et al. [9] presented a work concerned with the application of a turbine based on a nozzle named ACHEON (Aerial Coanda High Efficiency Orienting Nozzle). The studied configuration allowed to perform a selective adhesion of jet flows (in two streams) over two Coanda surfaces. Moreover, some theoretical recommendations about parameters as flight autonomy was obtained, energy consumption and forces acting in profiles evaluating the possibility of application of this main operational principle for design of airplanes.

However, in the naval industry the study of Coanda devices is underestimated in spite of several upgrades that can be made to improve efficiency in this field. During a journey, for example, rudders angles of attack fluctuate between -10° and 10° [22], causing an augmentation of propulsion power necessary to keep the ship velocity due to drag forces on the rudder. Then, the employment of nozzles to maneuverability can be an important application to minimize this kind of problem. The H.O.M.E.R. (High-Speed Orienting Momentum with Enhanced Reversibility) nozzle technology

has proven to be efficient for maneuverability of air vehicles [23]. Despite this new technology being proposed to aerospace vehicles, the main operational principle shall not be limited to air flows. In this context, the understanding of behavior of water flow in H.O.M.E.R nozzle is essential to verify its applicability for marine applications. A first attempt in this sense was performed in Lemos et al. [11], where a numerical study of turbulent water flows in two hydrodynamic profiles simulating the main operational principle of a hydro-propulsion device based on the Coanda effect was proposed. The influence of the distance between hydrodynamic profiles over mass flow rate, velocity and pressure fields were investigated and the results indicated a viability of this kind of main operational principle for water flows. Regardless of several important studies performed in the literature about turbulent flows in devices based on the Coanda effect, to the authors knowledge, only few studies considered water as working fluid, mainly in the H.O.M.E.R device. The present work is based on a proof of concept, i.e., if the main operational principle works for the present cases, it will have technical viability to be used for the design in marine applications for propulsion or maneuverability.

The present work aims to analyze numerically air and water flows in a H.O.M.E.R nozzle similar to that proposed by Trancossi and Dumas [24]. Both working fluids are considered incompressible. Moreover, turbulent flows at the steady state in a two-dimensional domain are simulated. The time-averaged conservation equations of mass and momentum are solved with the Finite Volume Method (FVM) [25–27]. To solve the closure problem of turbulence, the $k$-$\varepsilon$ model is employed [28,29]. Here, the numerical study is not concerned with the development of numerical methods for simulation of turbulent flows. However, important studies in the literature have been devoted for this kind of analysis [30,31]. In the present work, a first investigation of air flow in a H.O.M.E.R nozzle is simulated and results for deflection angle of jet are compared with those obtained by Trancossi and Dumas [24]. Afterwards, new theoretical recommendations about the fluid dynamic behavior of water flows in the device are obtained with the present numerical model. More precisely, the influence of the ratio between the mass flow rate in two inlets ($m^*$) over deflection angle of jet ($\alpha$) and velocity magnitudes of jet is evaluated. Moreover, the influence of $m^*$ on fluid dynamic behavior of flow in nozzle is investigated for two different septum insertions ($P$ = 10 mm and 48 mm).

## 2. Methodology

### 2.1. Description of the Problem

The study is performed through several numerical simulations using the FVM, more precisely with the commercial CFD (Computational Fluid Dynamic) software package FLUENT 14.0 [25]. The computational domain is similar to that presented by Trancossi and Dumas [24]. Figure 1a shows the whole computational domain of the problem and its respective boundary conditions, while Figure 1b shows a detailed nozzle view with its dimensions. Once two different magnitudes of variable $P$ = 10 mm and 48 mm (Figure 1b) are evaluated in the present work for air and water flows, this variable of septum insertion has no fixed magnitude in the sketch.

The H.O.M.E.R. nozzle is based on the Coanda effect. The main operational principle consists in forcing two fluid jets against two convex surfaces separated by a central septum (Figure 1a). Here, these two jets are imposed with velocities $V_1$ and $V_2$ in the upper and lower inlets, respectively. Those jets will adhere to the curved walls (where non-slip and impermeability boundary conditions are imposed) as a result of the differential pressure created by the viscous effects on the wall. This pressure gradient is mainly responsible for this attachment, once the ambient pressure is higher than the pressure near the wall. The end of the septum is characterized as a convergence zone of two imposed jets, starting the jets mixing process. The jet with the higher momentum will drag the other along, causing flow deflection. In the present simulations, the mixed jet flows in a region defined by a semi-circular area (gray area in Figure 1a) with dimensions $H = 2L = 1000$ mm. This domain is defined in such way that its external boundary conditions do not affect the flow near the nozzle. More precisely, atmospheric pressure in the exit lines of computational domain ($p_{man} = 0$ atm) is imposed.

Since the present work aims to analyze the behavior of flow in the nozzle using water as working fluid, a comparison of deflection angle of jet and velocity magnitudes in the *x* and *y* directions with air flows is made. In order to evaluate the flow deflection ($\alpha$), a total mass flow rate ($\dot{m}$) equals to 8.0 kg/s of water will be adopted. This magnitude is based on the study of Trancossi and Dumas [24] and represents values that can be found in real applications.

The deflection is a result of the difference between mass flow rate (which leads to difference of the momentum) of the flow in the superior and the inferior channels. This difference can be expressed in dimensionless form ($m*$) by the following expression [24]:

$$m* = \frac{\dot{m}_1 - \dot{m}_2}{\dot{m}_1 + \dot{m}_2} \tag{1}$$

here $\dot{m}_1$ and $\dot{m}_2$ are the mass flow rate in the superior and inferior channels, respectively. Those deflections will be measured with the software Digimizer [32], from the *x* axis in anticlockwise direction to the central region of the jet. However, the sum of $\dot{m}_1$ and $\dot{m}_2$ will always be equal to $\dot{m}$.

In addition, the magnitude velocity ($V$) is measured on the nozzle outlet, located tangentially to the two curved surfaces. Then, the *x* and *y* velocities ($V_x$ and $V_y$, respectively) are calculated through the following equations:

$$V_x = V * \cos \alpha \tag{2}$$

$$V_y = V * \sin \alpha \tag{3}$$

### 2.2. Mathematical and Numerical Modeling

The study deals with a turbulent, incompressible and steady state flow in a two-dimensional domain. The standard *k-ε* model (previously used in the study of Dumas and Trancossi [24]) was chosen to close the time-average equations. Considering this, the time-averaged conservation equations of mass and momentum are described by [29,33–35]:

$$\frac{\partial \overline{u}_i}{\partial x_i} = 0 \tag{4}$$

$$\frac{\partial}{\partial x_i}(\rho \overline{u}_i \overline{u}_j) = -\frac{\partial \overline{p}}{\partial x_j} + \frac{\partial}{\partial x_j}\left[ \mu \left( \frac{\partial \overline{u}_i}{\partial x_i} + \frac{\partial \overline{u}_j}{\partial x_i} \right) - \rho \overline{u'_i u'_j} \right] \tag{5}$$

where $u_i$ is the velocity in the *i*–direction, with *i* = 1 or 2 representing, respectively, the *x* and *y* direction. Moreover, *p* represents the pressure, $\rho$ is the fluid density, $\mu$ is the fluid viscosity, the overbar is the time average operator and $'$ represents the fluctuation fields of pressure and velocity.

The Reynolds stress can be related to the time-averaged deformation rate by:

$$\overline{u'_i u'_j} = \frac{\mu_T}{\rho}\left( \frac{\partial \overline{u}_i}{\partial x_j} + \frac{\partial \overline{u}_j}{\partial x_i} \right) - \frac{2}{3}k\delta_{ij} \tag{6}$$

where $\delta_{ij}$ is the Kronecker delta and the eddy viscosity ($\mu_T$) can be described, for the *k-ε* model, through the following equation [28]:

$$\mu_T = \rho C_\mu \frac{k^2}{\varepsilon} \tag{7}$$

where $C_\mu$ is a dimensionless constant (Table 1).

To obtain the turbulent kinetic energy (*k*) and its dissipation rate ($\varepsilon$) it is necessary to solve two additional transport equations given by:

$$\frac{\partial}{\partial x_i}(\rho u_i k) = \frac{\partial}{\partial x_i}\left[ \frac{\partial k}{\partial x_i}\left( \mu + \frac{\mu_t}{\sigma_k} \right) \right] + G_k - \rho \varepsilon - Y_M + \sigma_K \tag{8}$$

$$\frac{\partial}{\partial x_i}(\rho u_i \varepsilon) = \frac{\partial}{\partial x_i}\left[\frac{\partial \varepsilon}{\partial x_i}\left(\mu + \frac{\mu_t}{\sigma_\varepsilon}\right)\right] + C_{1\varepsilon}\frac{\varepsilon}{k}G_k - C_{2\varepsilon}\rho\frac{\varepsilon^2}{k} + \sigma_\varepsilon \tag{9}$$

where $G_k$ is the turbulent kinetic rate and $Y_M$ is the contribution of the turbulence to the total dissipation rate. The constants $C_{1\varepsilon}$, $C_{2\varepsilon}$, $\sigma_k$ and $\sigma_\varepsilon$ are shown in Table 1 [28].

Concerning the closure modeling, it is worth to mention that several important studies in the literature [35–37] have recommended the employment of the *k-ω* or SST *k-ω* models for simulation of shear flows as those found in external flows over cylinders, bluff bodies and jets. In spite of this recommendation, in the present work the *k-ε* model was used due to the fact that the results of Trancossi and Dumas [24] were achieved with this RANS closure model. Another important aspect is that the same cases simulated here with the *k-ε* model were repeated with the SST *k-ω* model and results were similar (with differences lower than 0.1%). These results are not presented in the present paper.

**Table 1.** Constants employed in the numerical model.

| $C_\mu$ | $C_{1\varepsilon}$ | $C_{2\varepsilon}$ | $\sigma_\varepsilon$ | $\sigma_k$ |
|---------|--------------------|--------------------|-----------------------|------------|
| 0.09 | 1.44 | 1.92 | 1.3 | 1.0 |

To solve the time-averaged equations and the additional transport equations presented, the FVM is used, more precisely, the Ansys Fluent CFD code (version 14.0) [25]. The solver is pressure based [25,38]. To treat advective terms, the Second Order Upwind advection scheme is utilized and for the pressure-velocity coupling the SIMPLEC method is applied. Residuals of $10^{-6}$ are employed for the conservation equations of the mass and momentum and the transport equations of $k$ and $\varepsilon$.

Concerning the spatial discretization, an illustration of the independent grid employed here is shown in Figure 2a. The independent grid is composed of 40,000 finite triangular and rectangular volumes. It is worthy to mention that 4 different meshes with 10,000, 20,000, 40,000 and 80,000 were simulated. For the sake of brevity, results for mesh independence test are not presented here. Moreover, 20 layers of inflation are inserted near the device walls to capture the turbulent boundary layer, as well as to stabilize quantities (as drag coefficient) at every iteration. A detailed view is shown in Figure 2b. Due to the lack of precision of the *k-ε* model to predict anisotropic gradients, the Enhanced Wall Treatment function was applied for the present simulations.

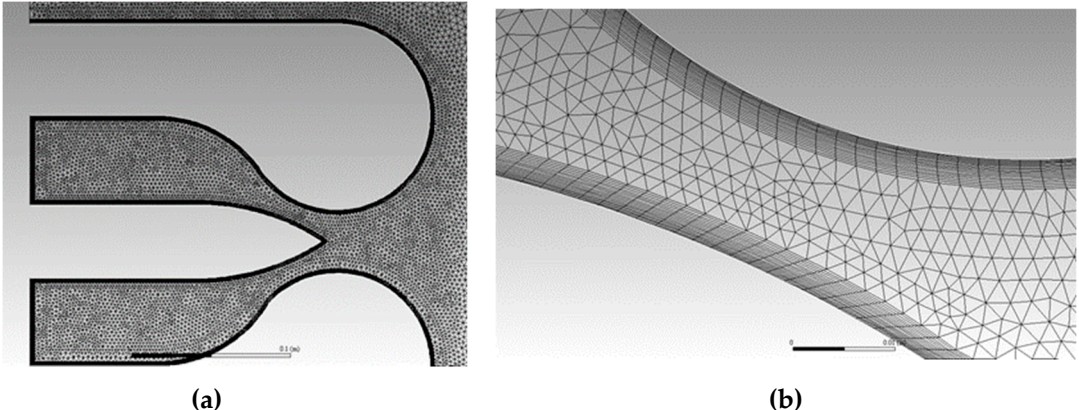

(a)　　　　　　　　　　　　(b)

**Figure 2.** Spatial discretization of the problem. (**a**) Overview of the mesh. (**b**) Detail of the mesh in the high momentum zone.

List of symbols and abbreviations can found in Appendix A.

## 3. Results and Discussion

### 3.1. Computational Model Verification

In order to evaluate the present numerical model, which is one of the purposes here, a verification is performed adopting the model established by Trancossi and Dumas [24]. Six different flow cases for each working fluid with different $m^*$ are simulated and the deflection angles ($\alpha$) are compared with those found by the authors. The magnitudes of $m^*$ adopted here and the mass flow rates injected in the superior and inferior channels are presented in Table 2.

**Table 2.** The $m^*$ chosen and mass flow rates imposed in each channel of the device.

| $m^*$ | $\dot{m}_1$ (kg/s) | $\dot{m}_2$ (kg/s) |
|---|---|---|
| 0.00 | 4.0 | 4.0 |
| 0.10 | 4.4 | 3.6 |
| 0.20 | 4.8 | 3.2 |
| 0.25 | 5.0 | 3.0 |
| 0.50 | 6.0 | 2.0 |
| 0.75 | 7.0 | 1.0 |

Figure 3 shows the comparison between the jet deflected angles ($\alpha$) as a function of dimensionless difference of mass flow rate ($m^*$) obtained with the present model and those predicted by Trancossi and Dumas [24]. Then, it can be observed that the results obtained in the present simulations have the same tendency predicted in the literature. For $m^* > 0.2$ it can be observed that the present results underestimate the deflection angle in comparison with numerical predictions of Trancossi and Dumas [24]. One possible explanation for the differences found here can be related with uncertainties introduced by two equations turbulence models, as has been reported in detail in literature reviews [39–41]. According to the authors, the RANS-based models have inherent inability to replicate fundamental turbulence processes, which leads to difficulties for the prediction of many base flows, mainly external flows and jets. Some difficulties for numerical prediction of jets with RANS models are properly presented in Mishra and Iaccarino [41]. In spite of some differences found for the work of Ref. [24] and some difficulties of RANS models for simulation of jet flows, the model employed here can be considered verified for the achievement of new theoretical recommendations for turbulent flows in the nozzle.

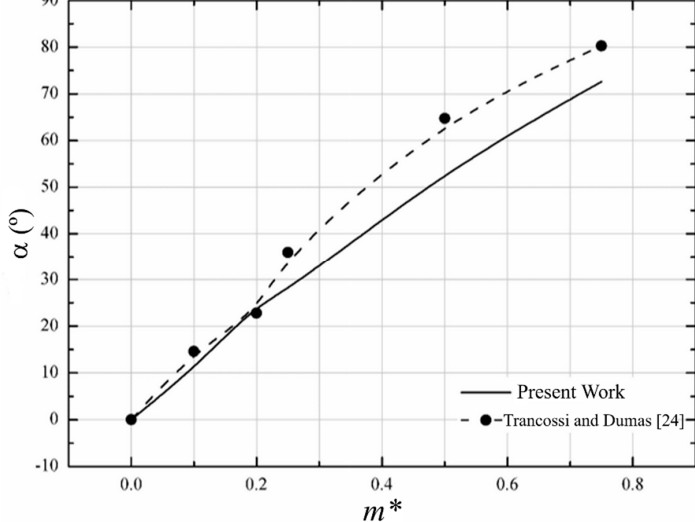

**Figure 3.** Comparison between the deflection angle of jet ($\alpha$) as a function of $m^*$ obtained in the present work and that predicted numerically by Trancossi and Dumas [24].

The next section is devoted to the simulation of water flows in the nozzle based on the Coanda effect considering two different septum insertion (*P*). More precisely, it is obtained the effect of *m** over deflection angle (*α*) and velocity magnitudes in *x* and *y* directions for water flows.

### 3.2. Results for Water Flows in the H.O.M.E.R. Nozzle

Initially, the influence of *m** over the deflection angles (*α*) for the two studied working fluids (air and water) is compared. Moreover, two different septum insertions were simulated (*P* = 10 mm and 48 mm) to investigate the influence of this variable on the deflection angle of air and water jets. Figure 4a,b show the results obtained injecting air and water in the device, respectively.

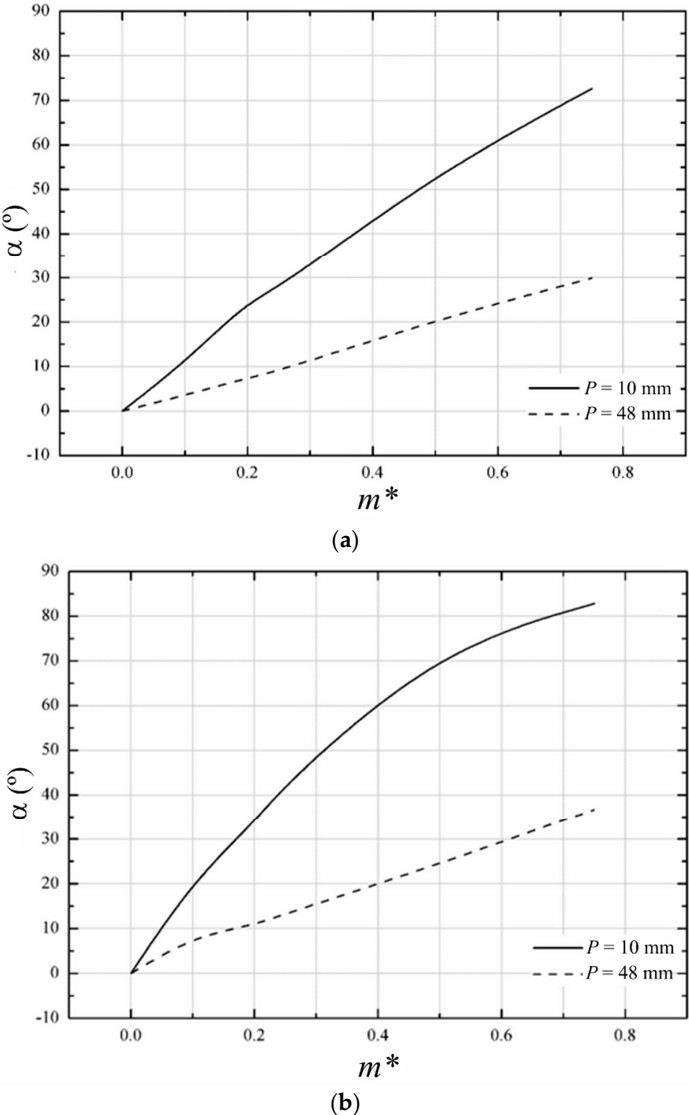

**Figure 4.** Deflection angles as function of *m** for different working fluids and two different septum insertions *P*. (**a**) Air (**b**) Water.

Figure 4a shows that the parameter *P* has a strong influence on the control of the deflection angle. For instance, an increase of nearly 2 times in magnitude of *α* for *P* = 10 mm in comparison with *P* = 48 mm is noticed. This increase is generated by a decrease of the channel height when the septum is inserted toward the convex walls. For *P* = 10 mm the fluid is forced to pass through a smaller space and it is accelerated. This increase in velocity leads to a decrease in absolute pressure, which raises the differential pressure between the flow and the ambient, and increases the space that the fluid remains

attached to the surface. Furthermore, a similar behavior can also be observed when water is simulated as working fluid, Figure 4b. It is also noticed that the device delivers larger deflection angles when water is used, with a deflection angle nearly 12% higher than that reached for air as working fluid. This can be explained in terms of inviscid irrotational flow, as stated by Bradshaw [25]. Adopting the flow as inviscid, the differential pressure is proportional to the specific mass of the fluid. As water has higher density than air, it will remain attached longer. Another interesting behavior is observed when the effect of $m^*$ over $\alpha$ is predicted for $P = 10$ mm with air and water. For air flow, the increase of $\alpha$ with $m^*$ is almost linear, while for water the increase is more intense for lower magnitudes of $m^*$ and with the augmentation of $m^*$ the variation of $\alpha$ decreases. This difference can be associated with difference of viscosity of the fluids, which for the present configurations affects the Reynolds number of fluid flow in each inlet.

Since the H.O.M.E.R. nozzle is a device designed to improve maneuverability, it is important to evaluate the effect of parameters $m^*$ and $P$ on the outlet velocity, because the velocity has a dominant effect on fluid dynamic forces. Thus, Figure 5; Figure 6 show the velocities in the $y$ and $x$ direction, respectively, as a function of $m^*$ for two different insertions of the central septum ($P$).

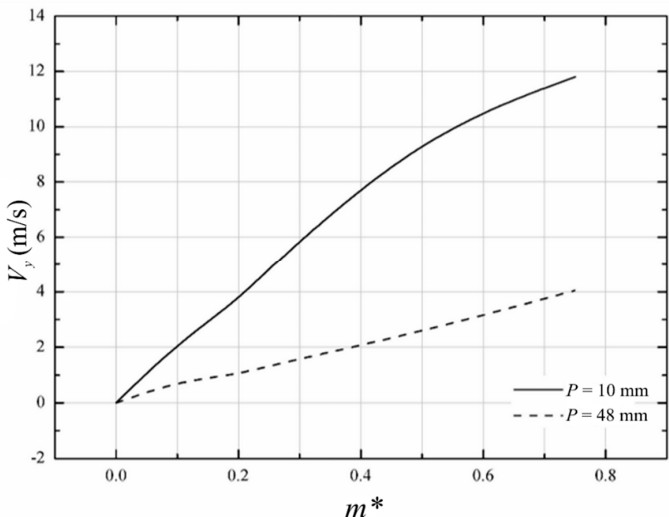

**Figure 5.** Effect of $m^*$ over magnitude of velocity in $y$ direction for two different values of $P$.

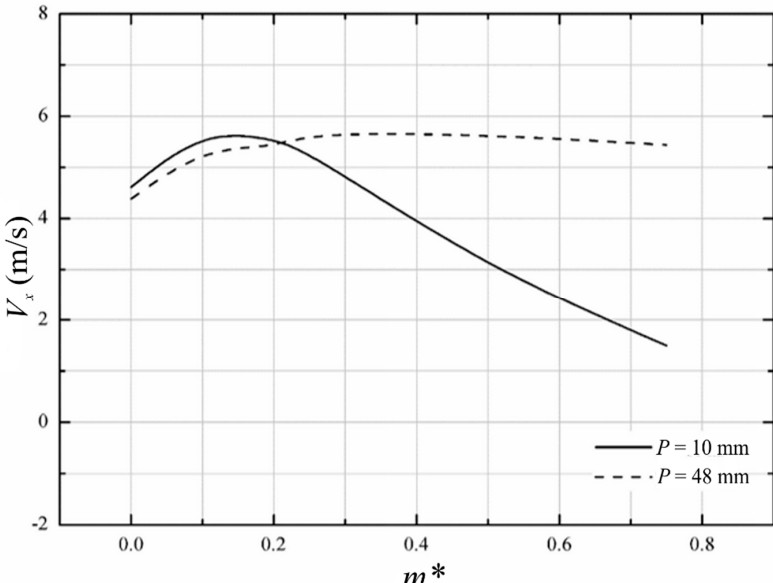

**Figure 6.** Effects of $m^*$ over magnitude of velocity in the $x$ direction for two different values of $P$.

Results indicated that the increase of *m*\* also led to an increase of magnitude of velocity in the *y* direction for both values of *P*, which is expected since the jet is deflecting towards the *y* axis. Results also showed that the lower magnitude of *P* led to a strong increase of *y* direction velocity. Concerning the velocity in the *x* direction, for *P* = 10 mm the opposite behavior is observed, i.e., the magnitude of the *x* velocity decreases with the increase of *m*\*. For *P* = 48 mm the magnitude of the *x* velocity is not much affected by variation of *m*\*. For lower magnitudes of *P*, results indicated that the strong restriction of channels increases the amount of momentum toward the *y* direction, but suppresses the momentum in the *x* direction. For the highest magnitude of *P*, the momentum magnitude is more equilibrated in both directions (with a tendency of increase for *y* direction velocity). In general, results indicated that this kind of main operational principle seems suitable for maneuverability of marine applications. Moreover, the ratio between mass flow rates in two inlet channels and dimension of central septum insertion can be used to control the deflection angle of the mixture jet and its intensity (momentum).

For a better visualization of the flow, velocity magnitude fields can be observed in Figure 7. In this image the variation of deflection angle of jet between the insertion *P* = 10 mm (Figure 7a) and *P* = 48 mm (Figure 7b) is clear. Fields of Figure 7 also show the increase of jet deflection towards the *y* axis with the increase of *m*\*. For *P* = 10 mm it can be seen that the jet is meagered when distant from the device as the magnitude of *m*\* increases. For *P* = 48 mm, the magnitude of jet velocity is high along a longer distance from the device, but the jet is more concentrated in the central region of domain, i.e., with lower deflection angles in comparison with those observed for *P* = 10 mm.

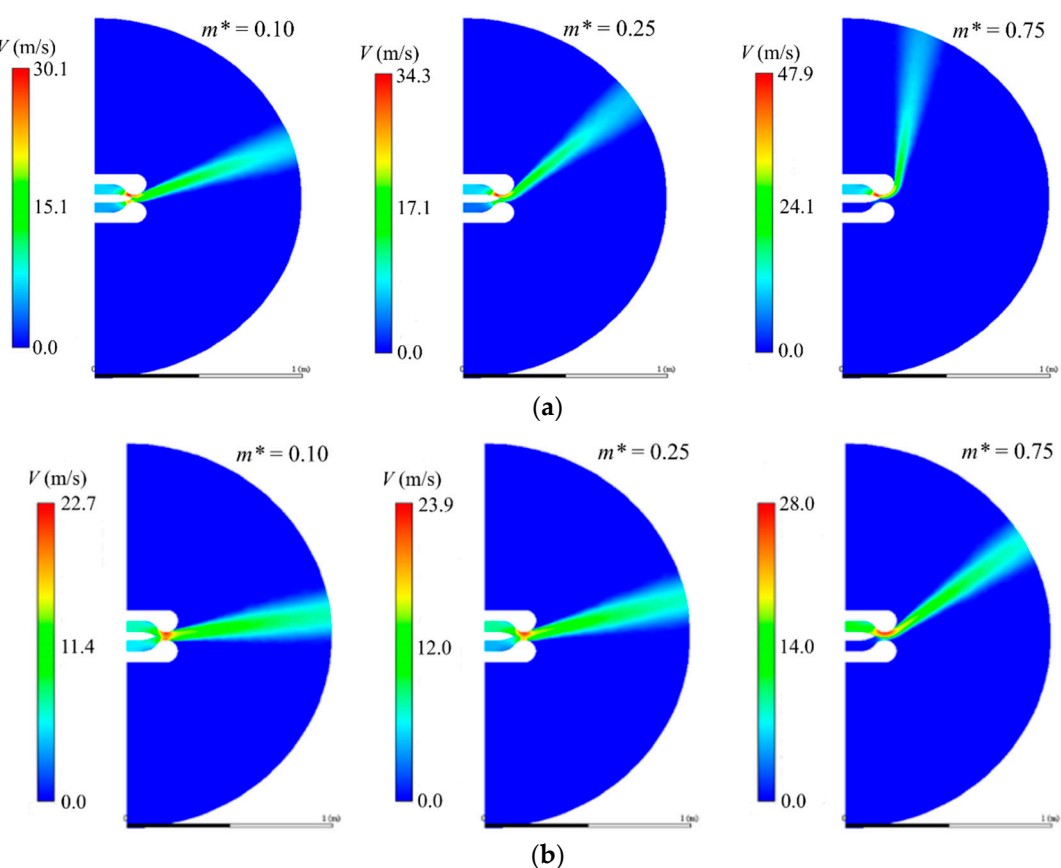

**Figure 7.** Velocity magnitude fields for different *m*\*: (**a**) *P* = 10 mm, (**b**) *P* = 48 mm.

As stated before, the differential pressure has a high influence on the detachment of a fluid jet on a curved surface. When the jet absolute pressure reaches the ambient pressure, the fluid will start

to detach from surface. That phenomenon can be noticed in Figure 8, where the fluid start to change direction at the same moment that the jet pressure becomes equal to the external pressure.

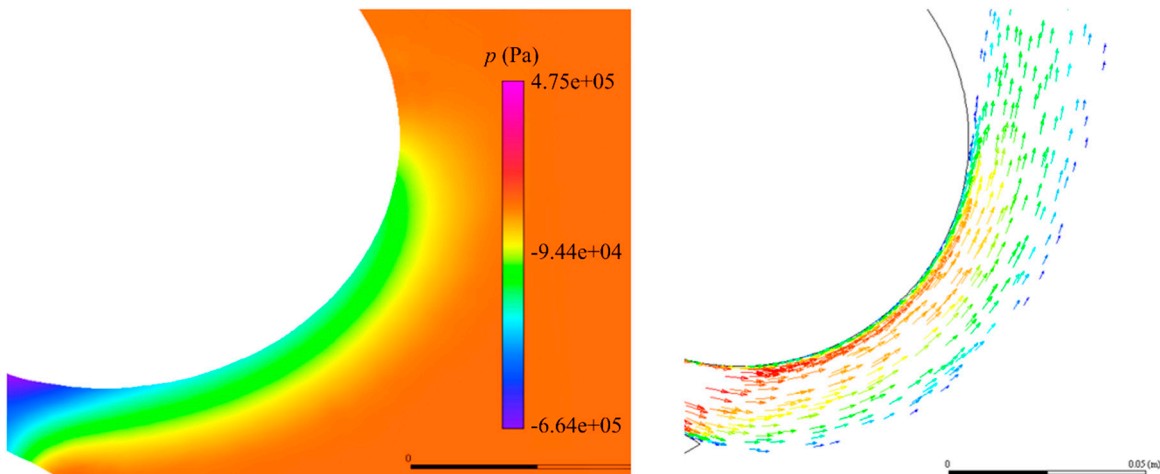

**Figure 8.** Absolute pressure field and vector velocity field for *P* = 10 mm and *m*\* = 0.75.

## 4. Conclusions

The present work showed promising results in relation to the use of water as working fluid in the H.O.M.E.R. nozzle. In general, the main operational principle based on the Coanda effect worked for water as working fluid. In this sense, the employment of this kind of main operational principle is technically feasible for the design of propulsion and/or maneuverability in marine applications. Results showed that the influence of the parameters *m*\* and *P* in the flow proved to be relevant, affecting the deflection angles magnitudes and velocities. When the insertion of the septum was equal to 10 mm, it shows angles 243% higher than the 48 mm insertion. However, the magnitude of velocity in the *x* direction suffered a strong reduction for higher magnitudes of *m*\*, which indicates that the jet flow is more concentrated near the nozzle and a reduction of the driven force can occur in the device. In other words, the two studied parameters can be used to control the jet deflection and magnitude in the present problem.

For future works, the evaluation of other magnitudes of *P* is recommended, as well as the study of forces acting in the device to obtain theoretical recommendations about the parameters that lead to the best performance of the device. Although the results proved to be favorable, an experimental model should be developed to validate the obtained numerical results.

## Authors Contributions

Conceptualization, C.H.M., R.d.L.L. and E.D.S.; Formal analysis, C.H.M. and R.d.L.L.; Funding acquisition, L.A.O.R., L.A.I. and E.D.S.; Investigation, Y.E.H. and R.d.L.L.; Methodology, Y.E.H., C.H.M., L.A.O.R., L.A.I. and R.d.L.L.; Project administration, E.D.S.; Resources, C.F.; Software, Y.E.H. and R.d.L.L.; Supervision, E.D.S.; Validation, Y.E.H. and R.d.L.L.; Visualization, C.F.; Writing—original draft, Y.E.H. and E.D.S.; Writing—review & editing, C.H.M., L.A.O.R., L.A.I. and C.F.

**Funding:** FAPERGS, Research grants of CNPq (Processes: 306024/2017-9, 306012/2017-0, 307847/2015-2) and Universal Project (Process: 445095/2014-8).

**Acknowledgments:** The author Y.E. Halal thanks FAPERGS by Scientific Initiation Scholarship. The author R.L. Lemos thanks CNPq for a Master Science Scholarship. The authors E.D. Santos, L.A. Isoldi and L.A.O. Rocha thank CNPq (Brasília, DF, Brazil) for the research Grant (Processes: 306024/2017-9, 306012/2017-0, 307847/2015-2) and for financial support in the Universal project (Process: 445095/2014-8).

**Conflicts of Interest:** The authors declare no conflict of interest.

## Appendix A

*Appendix A.1. List of Symbols*

| | |
|---|---|
| $H$ | Height of domain (mm) |
| $k$ | Turbulent kinetic energy (m$^2$ s$^{-2}$) |
| $L$ | Length of domain (mm) |
| $m^*$ | Dimensionless mass flow rate |
| $\dot{m}_1$ | Mass flow rate on the superior channel (kg s$^{-1}$) |
| $\dot{m}_2$ | Mass flow rate on the inferior channel (kg s$^{-1}$) |
| $P$ | Septum insertion distance (mm) |
| $p$ | Pressure (Pa) |
| $u_i$ | Velocity in the $i$–direction (m s$^{-1}$) |
| $V_1$ | Velocity at the inlet of the superior channel (m s$^{-1}$) |
| $V_2$ | Velocity at the inlet of the inferior channel (m s$^{-1}$) |
| $x_i$ | Spatial coordinate in the $i$–direction (m) |
| $'$ | Fluctuation fields of pressure and velocity |
| — | Time average operator |
| $\alpha$ | Deflection angle of the mixed jet (°) |
| $\delta_{ij}$ | Kronecker delta |
| $\varepsilon$ | Dissipation rate (m$^2$ s$^{-3}$) |
| $\mu$ | Dynamic viscosity (kg m$^{-1}$ s$^{-1}$) |
| $\mu_T$ | Turbulent eddy viscosity (kg m$^{-1}$ s$^{-1}$) |
| $\rho$ | Density (kg m$^{-3}$) |

*Appendix A.2. List of Abbreviations*

| | |
|---|---|
| ACHEON | Aerial Coanda High Efficiency Orienting Nozzle |
| CFD | Computational Fluid Dynamics |
| FVM | Finite Volume Method |
| H.O.M.E.R. | High-Speed Orienting Momentum with Enhanced Reversibility |
| SIMPLEC | Semi-Implicit Method for Pressure Linked Equations-Consistent algorithm |

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
