# Peer review of "Numerical Study of Turbulent Air and Water Flows in a Nozzle Based on the Coanda Effect"

_jmse, doi:10.3390/jmse7020021_

Round 1
Reviewer 1 Report
The present work performed a numerical study of turbulent flows in a H.O.M.E.R nozzle using water as working fluid, verifying whether the main operational principle (Coanda effect) works for a more viscous fluid as water, allowing a future application of this kind of device for maneuverability of marine applications. The time-averaged conservation equations of mass and momentum are solved with the finite volume method (FVM) and k – ε model is used to solve the closure problem.
I give my opinion on the suitability of the publication of this paper after the following revisions:
#1: In the Keywords should be taken new "Finite Volume Method (FVM)", or "FVM" only.
#2: The Nomenclature list has to be introduced at the end of the paper.
#3: Authors may explain deficiencies or shortcomings of other new studies to make a bridge to introducing the novelty of their work.
#4: Some important relevant references are missing, and they should be included in the background in the context of the paragraph: “In 1936, Henri Coanda patented the first device capable to deflect a stream without any movable parts [1]. This effect is named Coanda effect and consists on the tendency of a fluid to adhere to a curved surface due to the local pressure drop caused by acceleration of flow around a solid surface. Coanda Surfaces are generated by asymmetric profiles configured in such a way that the flow properties in the exit of devices are changed [2]. Thus, this phenomenon called Coanda Effect became the target of many studies where several mechanisms which work based on this effect were developed, such as cooling [3, 4, 5], safety devices [6], aerospace applications [7, 8, 9, 10] and few studies in marine devices [11, 1].”, should be noted importance of used Coanda Effect in UAV applications. Authors must discuss this point on the basis of the following new papers: |
B. M. Kulfan, Universal Parametric Geometry Representation Method, Journal of Aircraft, Vol. 45, No. 1, January–February 2008, pp. 142-158, doi:s10.2514/1.29958
Barlow Chris, Lewis Darren, Prior Stephen D., Odedra Sid, Erbil Mehmet Ali,Karamanoglu Mehmet and Collins, R., Investigating the use of the Coanda Effect to Create Novel Unmanned Aerial Vehicles, International Conference on Manufacturing and Engineering Systems, Proceedings . pp. 386-391, ISSN 2152-1522, 2009.
Mirkov, N., Rašuo, B., Numerical Simulation of Air Jet Attachment to Convex Walls and Applications, 27th ICAS Congress, 19 - 24 September, Nice, France, pp. 1–7 (CD-Rom), 2010
Mirkov, N., Rašuo, B., Maneuverability of an UAV With Coanda Effect Based Lift Production, 28th ICAS Congress, 23 - 28 September, Brisbane, Australia, pp. 1–6 (CD-Rom). 2012
Mirkov, N. and Rašuo, B., Numerical simulation of air jet attachment to convex walls and application to UAV, In: Knobloch, P. (Ed) Boundary and Interior Layers, Computational and Asymptotic Methods - BAIL 2014, Editor:, Series: Lecture Notes in Computational Science and Engineering, Vol. 108, Springer, pp. 197-208, 2015, doi: 10.1007/978-3-319-25727-3_15
#5: It is necessary in the context of the sentences: "The time-averaged conservation equations of mass and momentum are solved with the Finite Volume Method (FVM) [20 – 22]. To solve the closure problem of turbulence, the k – ε model is employed [23 – 24]. Firstly, air flow in a H.O.M.E.R nozzle is simulated and results for deflection angle of jet are compared with those obtained by Trancossi and Dumas [19].", very important to solve the closure problem of turbulence and should be discussed relying and on the work:
A. Pinelli, I. Naqavi, U. Piomelli, J. Favier, Immersed-boundary methods for general finite-difference and finite-volume Navier–Stokes solvers, Journal of Computational Physics, Vol. 229, No 24, pp. 9073–9091, December, 2010, doi:10.1016/j.jcp.2010.08.021
Mirkov, N., Rašuo, B. and Kenjereš, S.: On the improved finite volume procedure for simulation of turbulent flows over real complex terrains -
Journal of Computational Physics, Vol. 287, No 15, pp. 18-45, April 2015, doi: 10.1016/j.jcp.2015.02.001
#6: In the Conclusion, an abstract is repeated instead of a final note of the
results obtained in the paper.
The paper ID: jmse-399883, titled: “Numerical Study of Turbulent Air and Water Flows in a Nozzle Based on Coanda Effect”, from the authors: Youssef El Halal, Crístofer Marques, Luiz Rocha, Liércio Isoldi, Rafael Lemos, Cristiano Fragassa, Elizaldo dos Santos, is good, and I recommend it for publication in the Journal of Marine Science and Engineering, after suggested major revision.
Author Response
Reviewer 1
The present work performed a numerical study of turbulent flows in a H.O.M.E.R nozzle using water as working fluid, verifying whether the main operational principle (Coanda effect) works for a more viscous fluid as water, allowing a future application of this kind of device for maneuverability of marine applications. The time-averaged conservation equations of mass and momentum are solved with the finite volume method (FVM) and k – ε model is used to solve the closure problem.
I give my opinion on the suitability of the publication of this paper after the following revisions:
#1: In the Keywords should be taken new "Finite Volume Method (FVM)", or "FVM" only.
We thank the reviewer for the observation. We included the keyword in the text.
#2: The Nomenclature list has to be introduced at the end of the paper.
The nomenclature list was performed and introduced at the end of the paper.
#3: Authors may explain deficiencies or shortcomings of other new studies to make a bridge to introducing the novelty of their work.
The main literature about turbulent flows in nozzles based on Coanda Effect has been mainly devoted for simulation of air as working fluid. In this sense, most of applications where the effect has been studied is also related with air flow. In spite of several important contributions obtained in literature, at the authors knowledge, there is only few studies about turbulent flows in nozzles based on Coanda Effect considering more viscous fluids, as water. The present work is based on the proof of concept, i.e., if the main operational principle also works for water, it has technical feasibility to be used for future developments in marine applications. The present manuscript was revised to improve the explanation about this contribution.
#4: Some important relevant references are missing, and they should be included in the background in the context of the paragraph: “In 1936, Henri Coanda patented the first device capable to deflect a stream without any movable parts [1]. This effect is named Coanda effect and consists on the tendency of a fluid to adhere to a curved surface due to the local pressure drop caused by acceleration of flow around a solid surface. Coanda Surfaces are generated by asymmetric profiles configured in such a way that the flow properties in the exit of devices are changed [2]. Thus, this phenomenon called Coanda Effect became the target of many studies where several mechanisms which work based on this effect were developed, such as cooling [3, 4, 5], safety devices [6], aerospace applications [7, 8, 9, 10] and few studies in marine devices [11, 1].”, should be noted importance of used Coanda Effect in UAV applications. Authors must discuss this point on the basis of the following new papers:
- B. M. Kulfan, Universal Parametric Geometry Representation Method, Journal of Aircraft, Vol. 45, No. 1, January–February 2008, pp. 142-158, doi:s10.2514/1.29958
- Barlow Chris, Lewis Darren, Prior Stephen D., Odedra Sid, Erbil Mehmet Ali,Karamanoglu Mehmet and Collins, R., Investigating the use of the Coanda Effect to Create Novel Unmanned Aerial Vehicles, International Conference on Manufacturing and Engineering Systems, Proceedings . pp. 386-391, ISSN 2152-1522, 2009.
- Mirkov, N., Rašuo, B., Numerical Simulation of Air Jet Attachment to Convex Walls and Applications, 27th ICAS Congress, 19 - 24 September, Nice, France, pp. 1–7 (CD-Rom), 2010
- Mirkov, N., Rašuo, B., Maneuverability of an UAV With Coanda Effect Based Lift Production, 28th ICAS Congress, 23 - 28 September, Brisbane, Australia, pp. 1–6 (CD-Rom). 2012
- Mirkov, N. and Rašuo, B., Numerical simulation of air jet attachment to convex walls and application to UAV, In: Knobloch, P. (Ed) Boundary and Interior Layers, Computational and Asymptotic Methods - BAIL 2014, Editor:, Series: Lecture Notes in Computational Science and Engineering, Vol. 108, Springer, pp. 197-208, 2015, doi: 10.1007/978-3-319-25727-3_15
We thank the reviewer by references suggestion. The first paragraph of introduction section was changed to include the discussion about the importance of Coanda Effect for Unmanned Aerial Vehicles (UAV). Moreover, the above suggested references were inserted in the present manuscript.
#5: It is necessary in the context of the sentences: "The time-averaged conservation equations of mass and momentum are solved with the Finite Volume Method (FVM) [20 – 22]. To solve the closure problem of turbulence, the k – ε model is employed [23 – 24]. Firstly, air flow in a H.O.M.E.R nozzle is simulated and results for deflection angle of jet are compared with those obtained by Trancossi and Dumas [19].", very important to solve the closure problem of turbulence and should be discussed relying and on the work:
A. Pinelli, I. Naqavi, U. Piomelli, J. Favier, Immersed-boundary methods for general finite-difference and finite-volume Navier–Stokes solvers, Journal of Computational Physics, Vol. 229, No 24, pp. 9073–9091, December, 2010, doi:10.1016/j.jcp.2010.08.021
Mirkov, N., Rašuo, B. and Kenjereš, S.: On the improved finite volume procedure for simulation of turbulent flows over real complex terrains - Journal of Computational Physics, Vol. 287, No 15, pp. 18-45, April 2015, doi: 10.1016/j.jcp.2015.02.001
One additional sentence was intruded to explain the importance of studies in numerical methods and closure modeling for turbulent flows. The above recommended references were also included in the paper.
#6: In the Conclusion, an abstract is repeated instead of a final note of the results obtained in the paper.
The conclusion section was revised and changed in order to comply with the reviewer request.
The paper ID: jmse-399883, titled: “Numerical Study of Turbulent Air and Water Flows in a Nozzle Based on Coanda Effect”, from the authors: Youssef El Halal, Crístofer Marques, Luiz Rocha, Liércio Isoldi, Rafael Lemos, Cristiano Fragassa, Elizaldo dos Santos, is good, and I recommend it for publication in the Journal of Marine Science and Engineering, after suggested major revision.
We thank the reviewer again for the valuable considerations about our work.

Reviewer 2 Report
Please refer to required changes in the document that I am attaching.

Author Response
Reviewer 2
The authors carry out a numerical analysis of the flows of different fluids in a specific HOMER nozzle design using commercial CFD software. The overarching objective is to verify the applicability of the design for marine applications. The authors validate their numerical simulations via comparison with published results for air flows and extend the same for the water flows. The results of this article suggest that the HOMER nozzle designs can be used in marine applications and the advantages seen with air as a medium may be accrued for the marine flows as well.
While evaluating a manuscript for publication, there are two facets that must be considered:
- The quality of the presentation: this espouses the narrative being coherent and articulate, the text being largely free of typographical errors, the quality of the figures, no incorrect or un-supported claims being made, etc.
- The quality of the research: This includes the mathematical rigor and physics adherence of the steps, the depth of the analysis, the merit of the results, and, the utility of the investigation in advancing the state of knowledge in the sub-field.
With respect to the former, the text is rife with grammatical and typographical errors. I have pointed many of these out in Section III of my comments and ask the authors to correct them. Further, I urge the authors to proofread the manuscript carefully before re-submission. Such careless errors takes away from the quality of their work.
With respect to the latter criterion, I have some misgivings that I have pointed out in Section II of my comments. There are essential discussions and pertinent references that need to be added to ensure that the manuscript is publishable.
Owing to the importance and the quality of the results, I recommend this manuscript for publication in JMSE, once all the issues in Sections II and III of this report have been addressed.
We thank very much the reviewer by the valuable comments. All aspects pointed by reviewer were improved in the final version of the manuscript. A careful proofreading of the manuscript was performed to avoid the typo and grammatical errors seen in the first version of manuscript.
2. Omissions & Corrections
- Page 2: If the authors are using ANSYS Fluent for their CFD simulations, they must add the reference to the Fluent Solver. This would be given by: Fluent, A., 2009. 12.0 Theory Guide. Ansys Inc, 5(5). I would suggest that the authors include a small discussion of this and cite the reference.
The suggested reference was included in the manuscript and more details about the numerical solution was presented in Page 5 (Mathematical and Numerical Modeling Section).
- Page 4: The authors choose to use the k – ε turbulence model here. However, it has been shown in numerous studies that this model does not perform well for simulating turbulent jets12. Even at present, the k – ω model is preferred in the turbulence community for such flows. Could the authors justify their choice? I would suggest that the authors include a small discussion of this and cite the references.
1Morgans, R.C., Dally, B.B., Nathan, G.J., Lanspeary, P.V. and Fletcher, D.F., 1999, December. Application of the revised Wilcox (1998) k- turbulence model to a jet in co-flow. In Second International Conference on CFD in the Mineral and Process Industries, Melbourne, Australia.
2Georgiadis, Nicholas J., Dennis A. Yoder, and William B. Engblom. "Evaluation of modified two-equation turbulence models for jet flow predictions." AIAA journal 44, no. 12 (2006): 3107-3114.
We agree with the reviewer comments about the RANS closure models. In general, k – ε model is more recommended for isotropic flows, while k – ω is more suitable for anisotropic flows (as those achieved in near walls region and those with high shear rates as those found in external flows and jets). Then, for simulation of external flows, our research group has used in general the SST k – ω model, as can be seen in Teixeira et al. (2018). In spite of these recommendations in literature, the study of Trancossi and Dumas [24] used the k – ε model in their simulation. Once it is performed a comparison with the results of this work of Ref. [24] we decided to employ the same closure model.
Considering that the results for deflection angle (α) as function of m* reached here were not in strictly agreement with those of Ref. [24] all the simulations performed with the k – ε models were repeated by the authors with k – ω model. The results reached with both models were similar.
In order to comply with the reviewer request, an explanation about the closure modeling recommended in literature and that used here are inserted in the manuscript
Teixeira, F.B.; Lorenzini,G.; Errera, M.R.; Rocha, L.A.O.; Isoldi, L.A.; Dos Santos, E.D. Constructal Design of Triangular Arrangements of Square Bluff Bodies under Forced Convective Turbulent Flows. International Journal of Heat and Mass Transfer, 2018, 126, 521 – 535. https://doi.org/10.1016/j.ijheatmasstransfer.2018.04.134
- Page 5: The authors seem to state that for judging convergence, they only look at the convergence of the residuals by 6 orders of magnitude. However, this is hardly adequate. Did they also plot integrated quantities at every iteration like the coefficient of friction at the nozzle surfaces to ensure that these had reached constant values?
Yes, the integrated quantities as well as time averaged parameters were monitored at each iteration. The suitably convergence and stabilization of integrated quantities were possible only after the grid refinement near the solid walls.
- Page 5: The authors state that “The differences found can be associated with difficulties to reproduce strictly the same geometry adopted in the literature, since some geometric parameters were not specified”. However, this is not correct. Many investigators have shared detailed delinations of the HOMER nozzle structure and experimental data on the same3,4. I would suggest that the authors include a small discussion of this and cite the references.
The real rationale underlying the discrepancy between the validation data and the CFD predictions are the uncertainties introduced by the k – turbulence model. The uncertainties and errors in eddy-viscosity based model predictions for turbulent jet flows has been discussed in detail in literatures. I would suggest that the authors include a small discussion of this uncertainty and cite the reference.
3Trancossi, M., Stewart, J., Maharshi, S. and Angeli, D., 2016. Mathematical model of a constructal Coanda Effect nozzle. Journal of Applied Fluid Mechanics, 9(6), pp. 2813-2822.
4Subhash, M., Trancossi, M. and Pascoa, J., 2018. An Insight into the Coanda Flow Through Mathematical Modeling. In Modeling and Simulation in Industrial Engineering (pp. 101-114). Springer, Cham.
5Mishra, Aashwin Ananda, and Gianluca Iaccarino. "Uncertainty Estimation for Reynolds-Averaged Navier Stokes Predictions of High-Speed Aircraft Nozzle Jets." AIAA Journal (2017): 3999-4004.
We thank the reviewer for observation. We attempt to reproduce more accurately the simulations performed in literature. Once the present results were similar with two different closure models (k – ε and SST k – ω) we had assumed that differences could be associated with some differences in geometry. However, the explanation recommended by reviewer seems more elegant and reasonable. Then, we include the references and present explanation in the manuscript.
3 Grammatical & Typographical Errors
There are a lot of grammatical and typographical errors in the manuscript. I urge the authors to proofread the manuscript carefully before re-submission.
1. Page 2: “Despite this new technology is being proposed to aerospace vehicles” should be “Despite this new technology being proposed to aerospace vehicles”
We thank the reviewer by observation. The manuscript was thoroughly revised to correct these mistakes.
2. Page 2 “the understand of behavior of water flow in HOMER nozzle is essential to verify” should be “the understanding of behavior of water flow in HOMER nozzle is essential to verify”
This sentence and others were revised and corrected along the manuscript.
3. Page 4 “is the fluctuations of turbulent flows” should be “represents the fluctuating fields of pressure and velocity”
This sentence and others were revised and corrected along the manuscript.
4. Page 5 “results for grid independence test is not presented here.” Should be “results for grid independence test are not presented here”
This sentence and others were revised and corrected along the manuscript.
5. Page 6 “initially, it is compared the effect of m* over the deflection angles obtained with water and air.” Please re-express this sentence as it is incomprehensible to the reader.
This sentence and others were revised and corrected along the manuscript.
4. Summary & Recommendations
Owing to the importance and the quality of the results, I recommend this manuscript for publication in JMSE, once all the issues in Sections II and III of this report have been addressed.
We thank the reviewer again for the valuable comments and suggestions for improvement of the present manuscript.

Round 2
Reviewer 1 Report
The authors of the paper, ID: jmse-399883 - Revised Version, titled: " Numerical Study of Turbulent Air and Water Flows in a Nozzle Based on Coanda Effect", have respected and corrected everything I requested, so that I can accept it for publication without alternation.
Author Response
REPLY TO REVIEWERS' COMMENTS
We thank the reviewers for their interesting comments and suggestions. Changes in the manuscript are marked in red color. We respond below:
Editor and Reviewers
Both Reviewers are now happy how their technical points have been addressed. However, the grammatical and typographical error aspect mentioned by Reviewers 2 have not yet been resolved satisfactory. I believe it is in the interest of the Authors to publish an excellent contribution to promote their own special issue. The list below is aimed at assisting the Authors to improve these aspects:
L3: Many articles…ahead of i.
References: There are too many inconsistencies…and please write the Journal names in italic.
We thank the Reviewers and Editor for suggestions which significantly improved the manuscript.
All suggestions were implemented in the present manuscript. Figures were changed and References were also changed according to Reviewers and Editor recommendations.
